# A High-Precision Target Geolocation Algorithm for a Spaceborne Bistatic Interferometric Synthetic Aperture Radar System Based on an Improved Range–Doppler Model

Chao Xing [1], Zhenfang Li [1], Fanyi Tang [1,*], Feng Tian [2] and Zhiyong Suo [1]

[1] National Key Laboratory of Radar Signal Processing, Xidian University, Xi'an 710071, China; xingchao@xidian.edu.cn (C.X.); lzf@xidian.edu.cn (Z.L.); zysuo@xidian.edu.cn (Z.S.)
[2] Nanjing Research Institute of Electronics Technology, Nanjing 210039, China; feng_tian@stu.xidian.edu.cn
* Correspondence: 18021210100@stu.xidian.edu.cn; Tel.: +86-137-2073-0517

**Abstract:** A trend in the development of spaceborne Synthetic Aperture Radar (SAR) technology is the shift from a single-satellite repeated observation mode to a multi-satellite collaborative observation mode. However, current multi-satellite collaborative geolocation algorithms face challenges, such as geometric model mismatch and poor baseline estimation accuracy, arising from highly dynamic changes among multi-satellites. This paper introduces a high-precision and efficient geolocation algorithm for a spaceborne bistatic interferometric SAR (BiInSAR) system based on an improved range–Doppler (IRD) model. The proposed algorithm encompasses three key contributions. Firstly, a comprehensive description of the spatial baseline geometric model unique to the bistatic configuration is provided, with a specific focus on deriving the perpendicular baseline expression. Secondly, IRD geolocation functions are established to meet the specific requirements of the bistatic configuration. Then, a novel BiInSAR geolocation algorithm based on the IRD's functions is proposed, which can significantly improve the target geolocation accuracy by modifying the range–Doppler equation to suit the bistatic configuration. Meanwhile, a low-coupling parallel calculation method is proposed, which can improve the calculation speed by two to three times. Finally, the accuracy and efficiency of the algorithm are demonstrated using experimental data acquired by the TH-2 satellite, which is China's first spaceborne BiInSAR system. The experimental results prove that the IRD algorithm exhibits geolocation accuracy with an average error of less than 1 m and a standard deviation of less than 2.5 m while maintaining computational efficiency at a calculation speed of 1,429,678 pixels per second.

**Keywords:** interferometric synthetic aperture radar; InSAR; bistatic configuration; improved range–Doppler; IRD; geolocation algorithm; TH-2 satellite

## 1. Introduction

Spaceborne Interferometric Synthetic Aperture Radar (InSAR) has proven to be a highly effective method for acquiring high-resolution and high-precision digital elevation model (DEM) data [1–5]. The emergence of bistatic InSAR (BiInSAR) configurations, wherein two formation-flying satellites act as a single-pass radar interferometer, offers a novel and flexible approach to radar imaging, with distinct advantages over traditional repeat-pass interferometry [6,7]. Notable examples include the TerraSAR-X/TanDEM-X system developed by the German Aerospace Center [8–10], the TH-2 system developed by China [7,11], and the TwinSAR-L system developed by China [12–14].

However, the dynamic nature of the baseline in BiInSAR systems, owing to the rapid relative motion between satellites, presents a significant challenge. Existing baseline calibration models struggle to adequately describe the evolving baseline over time, necessitating innovative solutions. One such solution is the introduction of a pixel-related baseline model based on geometrical shifts [12], which accurately captures satellite position changes.

Furthermore, adapting repeat-pass InSAR geolocation algorithms to BiInSAR systems is non-trivial. A bistatic-to-monostatic equivalent imaging algorithm has been proposed to address this issue [15], simplifying the geolocation model by reducing the number of antenna phase centers (APCs) from two to one. However, this method assumes that the transmitter and receiver platforms follow a rectilinear path with constant and equal velocities, a condition not met by all BiInSAR systems. Researchers have consequently explored direct bistatic imaging algorithms capable of broader applications [16–19]. For instance, a three-dimensional localization approach based on the numerical range–Doppler algorithm and entropy minimization principle was introduced in [20], enhancing image quality and localization accuracy. Nonetheless, this approach does not fully exploit phase information within the images and overlooks the impact of system errors such as atmospheric delays and baseline measurement errors.

While bistatic-to-monostatic equivalent algorithms simplify the bistatic geolocation model, their applicability is constrained [15,21]. Direct bistatic geolocation algorithms, although more accurate, often demand time-consuming iterative operations [20,22]. Therefore, this paper concentrates on enhancing both the operational efficiency and accuracy of direct bistatic geolocation.

First, an improved range–Doppler (IRD) model is proposed to solve the geometric model mismatch problem in the BiInSAR system. The IRD model fully considers the different propagation paths between the transmitting antenna, receiving antenna, and ground targets. Then, an accurate target geolocation equation based on the IRD model is established. It can overcome the slant-range measurement error caused by APC equivalence and the Doppler frequency error caused by velocity equivalence in conventional geolocation algorithms. Apart from the efforts to improve geolocation accuracy, the challenge of accelerating calculation efficiency is another issue that researchers are interested in [23]. In this paper, a low-coupling parallel calculation method is proposed, which can improve the calculation speed by two to three times.

Second, a bistatic interferometric baseline calibration method based on an unwrapped interferogram and ground control points (GCPs) is introduced. The interferometric baseline vector is dissected into two orthogonal components: the parallel baseline, aligned with the line of sight (LoS), and the perpendicular baseline, orthogonal to the LoS. The measurement error of the parallel baseline can be estimated according to the linear relationship between the elevation measurement error and the parallel baseline measurement error. The perpendicular baseline can be determined via the least-squares fitting method. When constructing the baseline error-solving equation, the differences between the BiInSAR geometry and monostatic InSAR (MoInSAR) geometry are considered, and the information contained in the interference phase is fully utilized.

The rest of this article is organized as follows. Section 2 presents the geometry model of the bistatic interferometric baseline. The IRD model used for geolocation is presented, and its influencing factors are analyzed in Section 3. Section 4 introduces the implementation process of the algorithm, including the ICESat-2 data filtering method, baseline calibration method, IRD geolocation method, and parallel processing method. The accuracy and efficiency of the proposed algorithm are verified through two groups of experiments in Section 5. Finally, Section 6 summarizes the conclusions.

## 2. Interferometric Baseline Calibration

At the beginning of this section, the geometry model of the BiInSAR baseline configuration is described. Then, the principle of the baseline projection method from a three-dimensional vector to a two-dimensional vector is explained, and the mathematical expression of the baseline is deduced.

### 2.1. Geometry Model of BiInSAR Baseline Configuration

As shown in Figure 1a, two satellites are flying in approximately parallel trajectories. The trajectories of the active satellite $S_{AT}$ and passive satellite $S_{PR}$ are represented by the

red dotted line and the blue dotted line, respectively. An italicized $S_{AT}/S_{PR}$ denotes the satellite's name, whereas a bold $\mathbf{S_{AT}}/\mathbf{S_{PR}}$ represents the satellite's position vectors. The red and blue arrows represent the flight directions of $S_{AT}$ and $S_{PR}$, respectively. The yellow line between $\mathbf{S}_{AT}$ and $\mathbf{S}_{PR}$ represents the instantaneous baseline (ISB). In Figure 1b, the green line between $\mathbf{S}_{AT}$ and $\mathbf{S}_{PR}$ represents the interferometric baseline (ITB). The ISB in Figure 1a reflects the relative position relationship between the two satellites at the same time, whereas the ITB in Figure 1b represents the relative position of $S_{AT}$ and $S_{PR}$ at different moments but with the same view. The yellow triangle $P_T$ at the bottom of Figure 1b symbolizes an arbitrary target on Earth's surface. In Figure 1b, the red dashed line from $S_{AT}$ to $P_T$ indicates the signal transmission propagation path $R_{AT}$, whereas the blue dashed line from $P_T$ to $S_{PR}$ signifies the signal reception propagation path $R_{PR}$. Similarly, $S_{AR}$'s receiving signal propagation path $R_{AR}$ is represented by a red dashed line extending from $P_T$ to $S_{AR}$.

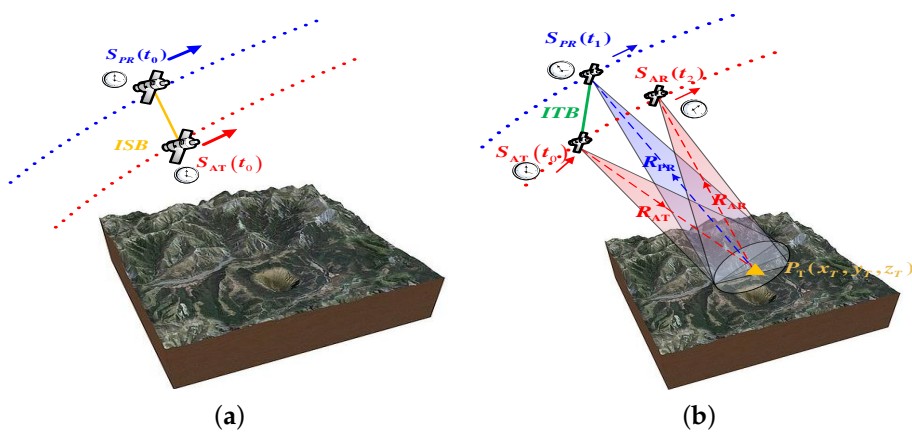

(**a**)  (**b**)

**Figure 1.** Geometry model of BiInSAR baseline configuration. (**a**) ISB at the same moment. (**b**) ITB at different moments with the same view.

The ITB cannot be measured directly since the differential GPS device can only measure the relative position relationship at the same moment, and SAR image products do not contain prior information about when $S_{AT}$ and $S_{PR}$ have the same view. Fortunately, the moments of the same view can be estimated from BiInSAR imaging geometry. Firstly, a ground target with three-dimensional coordinates is extracted from reference the DEM according to the latitude range and longitude range of the observed scene. The symbol $\mathbf{P}_T=(x_T,y_T,z_T)$ represents the position of a ground target. Then, the zero-Doppler moments of $S_{AT}$ and $S_{PR}$ can be searched according to Equations (1) and (2).

$$f_{d_{PR}}(t_0;t_1) = -\frac{\mathbf{V}_{AT}(t_0) \cdot (\mathbf{S}_{AT}(t_0) - \mathbf{P}_T)}{\lambda \|\mathbf{S}_{AT}(t_0) - \mathbf{P}_T\|} - \frac{\mathbf{V}_{PR}(t_1) \cdot (\mathbf{S}_{PR}(t_1) - \mathbf{P}_T)}{\lambda \|\mathbf{S}_{PR}(t_1) - \mathbf{P}_T\|} \tag{1}$$

$$f_{d_{AT}}(t_0;t_2) = -\frac{\mathbf{V}_{AT}(t_0) \cdot (\mathbf{S}_{AT}(t_0) - \mathbf{P}_T)}{\lambda \|\mathbf{S}_{AT}(t_0) - \mathbf{P}_T\|} - \frac{\mathbf{V}_{AT}(t_2) \cdot (\mathbf{S}_{AT}(t_2) - \mathbf{P}_T)}{\lambda \|\mathbf{S}_{AT}(t_2) - \mathbf{P}_T\|} \tag{2}$$

where $f_{d_{PR}}$ and $f_{d_{AT}}$ represent the Doppler frequency of a dual-satellite cooperative operation and the Doppler frequency of a single-satellite independent operation, respectively. The symbol $\lambda$ represents the carrier wavelength of the radar signal.

Geometric registration methods are usually used to calculate when two satellites have the same view [24], and the obtained time is called the Doppler center time (DCT). Then, the APC positions of satellite $S_{AT}$ and satellite $S_{PR}$ can be acquired using interpolation functions based on the original orbit information and DCT.

## 2.2. Projection Principle of BiInSAR Baseline

This subsection introduces the projection principle of the **ITB** from a three-dimensional vector to a two-dimensional vector. As depicted in Figure 2a, the **ITB** vector is pointing

from $S_{AT}$ at time $t_0$ to $S_{PR}$ at time $t_1$. The mathematical expression of the **ITB** can be expressed by Equation (3).

$$\mathbf{ITB}(t_0, t_1) = \mathbf{S}_{AT}(t_0) - \mathbf{S}_{PR}(t_1) \tag{3}$$

The **ITB** can be dissected into two orthogonal components: the across-track baseline **ACB** and the along-track baseline **ALB**. The **ALB** is parallel to satellite velocity $\mathbf{V}_{AT}$, and the **ACB** is perpendicular to $\mathbf{V}_{AT}$. The **ALB** is mainly used for ground moving-target velocity estimation, and the **ACB** is mainly used for topography mapping and ground deformation detection applications. The focus of this paper lies in the field of topography mapping, thus necessitating a meticulous estimation of the **ACB**. The green plane represents the zero-Doppler plane (ZDP) of a ground target $P_T$. Because both the **ACB** and ZDP are perpendicular to $\mathbf{V}_{AT}$, the **ACB** can be deemed as the projection of the **ITB** in a two-dimensional plane. The red point $S_{AP}$ is the projection of $S_{AT}$ in the ZDP. Similarly, the blue point $S_{PP}$ is the projection of $S_{PR}$ in the ZDP. The existing BiInSAR baseline calibration algorithms are all defined in a two-dimensional plane, called the ZDP, as shown in Figure 2b. The mathematical formulation presented in this study establishes a conversion relationship between the **ITB** and **ACB**.

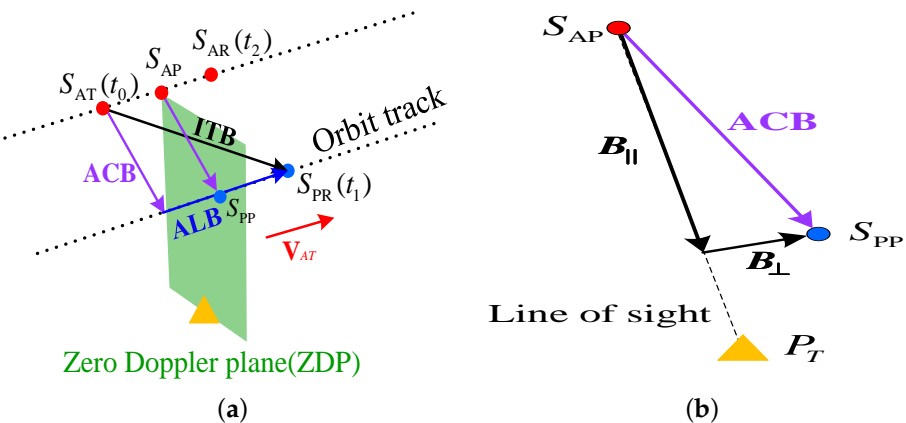

**Figure 2.** Projection principle of BiInSAR baseline. (**a**) Three-dimensional baseline vector. (**b**) Two-dimensional baseline vector.

It can be seen that the **ACB** is perpendicular to $\mathbf{V}_{AT}$, but this is not enough to determine its specific direction in the ZDP. The **ALB** vector is computed first, followed by the **ACB** based on the **ALB**. The **ALB** is parallel to $\mathbf{V}_{AT}$, and the modulus of the **ALB** can be calculated through a vector dot operation, as shown in Equation (4).

$$\|\mathbf{ALB}(t_0, t_1)\| = \mathbf{ITB}(t_0, t_1) \cdot \mathbf{V}_{AT}(t_0) / \|\mathbf{V}_{AT}(t_0)\| \tag{4}$$

In Equation (4), the symbol $\|\cdot\|$ represents the modulus of a vector. $\mathbf{V}_{AT}(t_0) / \|\mathbf{V}_{AT}(t_0)\|$ represents the unit vector along $\mathbf{V}_{AT}$ at time $t_0$. Then, the **ACB** can be calculated through a vector subtraction operation, as shown in Equation (5).

$$\mathbf{ACB}(t_0, t_1) = \mathbf{ITB}(t_0, t_1) - \mathbf{ALB}(t_0, t_1) \tag{5}$$

In Figure 2b, the $\mathbf{B}_\|$ component is pointing from $\mathbf{S}_{AP}$ to $\mathbf{P}_T$ along the radar line of sight (LoS). $\mathbf{B}_\perp$ is pointing from $\mathbf{S}_{PP}$ to the LoS, and it is perpendicular to the LoS. The direction of the LoS varies from near range to far range, so the directions of $\mathbf{B}_\|$ and $\mathbf{B}_\perp$ are also dynamic.

## 3. The IRD Model and Analysis of Influencing Factors

In this section, the IRD model is proposed, which can effectively reflect the geometric characteristics in the bistatic configuration, and the influencing factors of geolocation accuracy are analyzed.

### 3.1. Introduction to the Proposed IRD Model

This subsection introduces the proposed IRD geolocation model. First, an equation system for target position $\mathbf{P}_T$ is given according to the slant-range information and Doppler information, as shown in Equations (6)–(8). Equation (6) expresses the slant-range relationship between $\mathbf{P}_T$ and $\mathbf{S}_{AT}/\mathbf{S}_{AR}$. $\mathbf{P}_T$ is distributed over an ellipse, with $\mathbf{S}_{AT}$ and $\mathbf{S}_{AR}$ as the focal points, as shown in Figure 3. The solid red line represents the slant range between the active satellite and target, containing $R_{AT}$ and $R_{AR}$. $R_{AT}$ represents the distance from $\mathbf{S}_{AT}$ to $\mathbf{P}_T$, and $R_{AR}$ represents the distance from $\mathbf{P}_T$ to $\mathbf{S}_{AR}$. In Equation (7), $R_{PR}$ represents the distance from $\mathbf{P}_T$ to $\mathbf{S}_{PR}$. There are two methods used to calculate $R_{PR}$. One is based on the delay time from signal transmission to signal reception and is usually used in stereo SAR geolocation applications. The other is based on the interferometry phase (ITP), which is adopted in the InSAR geolocation algorithm. In Equation (8), $f_{dc}$ represents the Doppler center frequencies of $\mathbf{S}_{AT}$ and $\mathbf{S}_{AR}$ at target $\mathbf{P}_T$. The conversion relationship between the ITP and $R_{PR}$ can be expressed by Equation (9). In Equation (9), $\phi$ represents the ITP.

$$f_{RA}(\mathbf{S}_{AT}(t_0), \mathbf{S}_{AR}(t_2)) = \|\mathbf{S}_{AT}(t_0) - \mathbf{P}_T\| + \|\mathbf{S}_{AR}(t_2) - \mathbf{P}_T\| - R_{AT} - R_{AR} = 0 \quad (6)$$

$$f_{RP}(\mathbf{S}_{AT}(t_0), \mathbf{S}_{PR}(t_1)) = \|\mathbf{S}_{AT}(t_0) - \mathbf{P}_T\| + \|\mathbf{S}_{PR}(t_1) - \mathbf{P}_T\| - R_{AT} - R_{PR} = 0 \quad (7)$$

$$f_{DA}(\mathbf{S}_{AT}(t_0), \mathbf{S}_{AR}(t_2), \mathbf{P}_T) = -\frac{\mathbf{V}_{AT}(t_0) \cdot (\mathbf{S}_{AT}(t_0) - \mathbf{P}_T)}{\lambda \cdot \|\mathbf{S}_{AT}(t_0) - \mathbf{P}_T\|} - \frac{\mathbf{V}_{AR}(t_2) \cdot (\mathbf{S}_{AR}(t_2) - \mathbf{P}_T)}{\lambda \cdot \|\mathbf{S}_{AR}(t_2) - \mathbf{P}_T\|} - f_{dc} = 0 \quad (8)$$

$$R_{PR} = R_{AR} + \frac{\phi}{2\pi}\lambda \quad (9)$$

where $\mathbf{S}_{AT} = (x_{AT}, y_{AT}, z_{AT})$, $\mathbf{S}_{AR} = (x_{AR}, y_{AR}, z_{AR})$, $\mathbf{S}_{PR} = (x_{PR}, y_{PR}, z_{PR})$, $\mathbf{V}_{AT} = (v_{xt}, v_{yt}, v_{zt})$, and $\mathbf{V}_{AR} = (v_{xr}, v_{yr}, v_{zr})$.

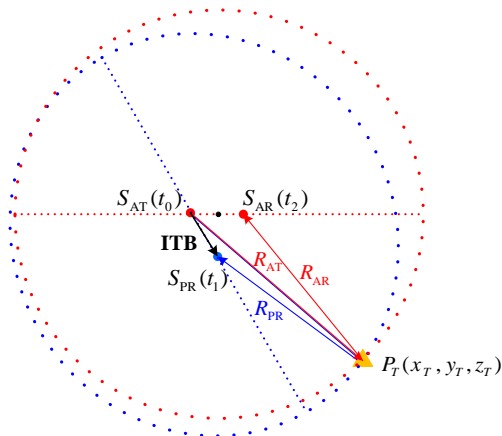

**Figure 3.** The proposed IRD model.

Considering there is an addition operation of two square roots in Equations (6) and (7), it is impossible to calculate the coordinate value of $\mathbf{P}_T$ through a closed-form expression. The Newton iteration method is a commonly used method to solve such nonlinear equations. Subsequently, a comprehensive explanation of the process for deriving the formula is provided. In Equation (10), to obtain the target geolocation sensitivity of the $f_{RA}$ range equation in the x-axis, the partial derivative of $f_{RA}$ with respect to $x_T$ is calculated.

Then, the partial derivatives of $f_{RA}$ with respect to $y_T$ and $z_T$ are calculated to obtain the sensitivity of the range equation in the y-axis and z-axis, as seen in Equations (11) and (12).

$$\frac{\partial f_{RA}}{\partial x_T} = -\frac{x_{AT}-x_T}{R_{AT}} - \frac{x_{AR}-x_T}{R_{AR}} \tag{10}$$

$$\frac{\partial f_{RA}}{\partial y_T} = -\frac{y_{AT}-y_T}{R_{AT}} - \frac{y_{AR}-y_T}{R_{AR}} \tag{11}$$

$$\frac{\partial f_{RA}}{\partial z_T} = -\frac{z_{AT}-z_T}{R_{AT}} - \frac{z_{AR}-z_T}{R_{AR}} \tag{12}$$

Similarly, the target geolocation sensitivity of the $f_{RP}$ range equation in three axes can be obtained by performing a partial derivative operation, as shown in Equations (13)–(15).

$$\frac{\partial f_{RP}}{\partial x_T} = -\frac{x_{AT}-x_T}{R_{AT}} - \frac{x_{PR}-x_T}{R_{PR}} \tag{13}$$

$$\frac{\partial f_{RP}}{\partial y_T} = -\frac{y_{AT}-y_T}{R_{AT}} - \frac{y_{PR}-y_T}{R_{PR}} \tag{14}$$

$$\frac{\partial f_{RP}}{\partial z_T} = -\frac{z_{AT}-z_T}{R_{AT}} - \frac{z_{PR}-z_T}{R_{PR}} \tag{15}$$

To solve the IRD equation system, it is also necessary to perform a partial derivative operation of $f_{DA}$ with respect to $x_T$, $y_T$, and $z_T$, as shown in Equations (16)–(18).

$$\frac{\partial f_{DA}}{\partial x_T} = \frac{-v_{xt}\cdot(x_T-x_{AT})R_{AT}^2+(x_T-x_{AT})\cdot \mathbf{V}_{AT}\cdot(\mathbf{P}_T-\mathbf{S}_{AT})}{\lambda R_{AT}^3}$$
$$+ \frac{-v_{xr}\cdot(x_T-x_{AR})R_{AR}^2+(x_T-x_{AR})\cdot \mathbf{V}_{AR}\cdot(\mathbf{P}_T-\mathbf{S}_{AR})}{\lambda R_{AR}^3} \tag{16}$$

$$\frac{\partial f_{DA}}{\partial y_T} = \frac{-v_{yt}\cdot(y_T-x_{AT})R_{AT}^2+(y_T-y_{AT})\cdot \mathbf{V}_{AT}\cdot(\mathbf{P}_T-\mathbf{S}_{AT})}{\lambda R_{AT}^3}$$
$$+ \frac{-v_{yr}\cdot(y_T-y_{AR})R_{AR}^2+(y_T-y_{AR})\cdot \mathbf{V}_{AR}\cdot(\mathbf{P}_T-\mathbf{S}_{AR})}{\lambda R_{AR}^3} \tag{17}$$

$$\frac{\partial f_{DA}}{\partial z_T} = \frac{-v_{zt}\cdot(z_T-z_{AT})R_{AT}^2+(z_T-z_{AT})\cdot \mathbf{V}_{AT}\cdot(\mathbf{P}_T-\mathbf{S}_{AT})}{\lambda R_{AT}^3}$$
$$+ \frac{-v_{zr}\cdot(z_T-z_{AR})R_{AR}^2+(z_T-z_{AR})\cdot \mathbf{V}_{AR}\cdot(\mathbf{P}_T-\mathbf{S}_{AR})}{\lambda R_{AR}^3} \tag{18}$$

By combining Equations (10)–(18), the iterative solution formula of target coordinates $(x_T, y_T, z_T)^T$ is obtained.

$$\begin{bmatrix} \frac{\partial f_{RA}}{\partial x_T}\big|_{x_T=x_T^k} & \frac{\partial f_{RA}}{\partial y_T}\big|_{y_T=y_T^k} & \frac{\partial f_{RA}}{\partial z_T}\big|_{z_T=z_T^k} \\ \frac{\partial f_{RP}}{\partial x_T}\big|_{x_T=x_T^k} & \frac{\partial f_{RP}}{\partial y_T}\big|_{y_T=y_T^k} & \frac{\partial f_{RP}}{\partial z_T}\big|_{z_T=z_T^k} \\ \frac{\partial f_{DA}}{\partial x_T}\big|_{x_T=x_T^k} & \frac{\partial f_{DA}}{\partial y_T}\big|_{y_T=y_T^k} & \frac{\partial f_{DA}}{\partial z_T}\big|_{z_T=z_T^k} \end{bmatrix} \cdot \begin{bmatrix} \Delta x_T^{k+1} \\ \Delta y_T^{k+1} \\ \Delta z_T^{k+1} \end{bmatrix} = \begin{bmatrix} \Delta R_A \\ \Delta R_P \\ \Delta f_d \end{bmatrix} \tag{19}$$

where $P_T^k = (x_T^k, y_T^k, z_T^k)^T$ represents the current iteration value, and $P_T^{k+1} = (x_T^{k+1}, y_T^{k+1}, z_T^{k+1})^T$ represents the next iteration value. When $k = 0$, $P_T^k$ represents the initial value of the iteration. $\mathbf{e} = (\Delta R_A, \Delta R_P, \Delta f_D)^T$ represents the geolocation error estimation value of the current observation vector. $\mathbf{u} = \left(\Delta x_T^{k+1}, \Delta y_T^{k+1}, \Delta z_T^{k+1}\right)^T$ represents the iterative correction value. Equation (19) can be expressed in matrix form as follows:

$$\mathbf{A} \cdot \mathbf{u} = \mathbf{e} \tag{20}$$

### *3.2. Influencing Factors of Geolocation Accuracy*

There are two types of factors influencing geolocation accuracy: the measurement errors of radar parameters and the errors introduced by InSAR signal processing [25]. As illustrated in Figure 4, the measurement errors include the satellite timing error, APC position vector measurement error, and APC velocity vector measurement error [26]. The errors introduced by InSAR signal processing include the interferometric baseline estimation error [27], atmospheric delay estimation error [28], and interferogram phase unwrapping error [29].

The satellite timing error affects the records of image start and end times, thus indirectly influencing the interpolation result of satellite orbital state vectors [26]. This timing error is a kind of system error, and it can be estimated and corrected by inter-satellite time synchronization signals. The measurement values of APC position and velocity vectors are extracted from satellite ephemeris data, and different kinds of ephemeris data have different accuracies. Ephemeris data can be classified into three levels: precise ephemeris data, fast ephemeris data, and real-time ephemeris data [30]. Precise ephemeris data are the most precise but are not available until 12 days after the SAR image is acquired. Fast ephemeris data are moderately accurate and are available 2 days after the SAR image is acquired. Real-time ephemeris data are less accurate but are available as soon as the SAR image is acquired. So, the measurement errors of APC position and velocity vectors can be reduced using high-precision ephemeris data.

The interferometric baseline error is a kind of system error, and it influences the position vector accuracy of $\mathbf{S}_{PR}$. The GCPs or ICESat-2 laser altimetry data can serve as a valuable reference for calibrating the interferometric baseline error. The atmospheric delay error is a kind of random error, and it is closely related to meteorological conditions such as cloud thickness and water vapor content [28]. The atmospheric delay error can be estimated using statistical model methods or meteorological data methods [31]. The interferometric phase unwrapping error is a kind of random error, and it influences the range accuracy of $\hat{R}_{ATP}$. The interferometric phase unwrapping error can be reduced using a multi-baseline phase unwrapping algorithm.

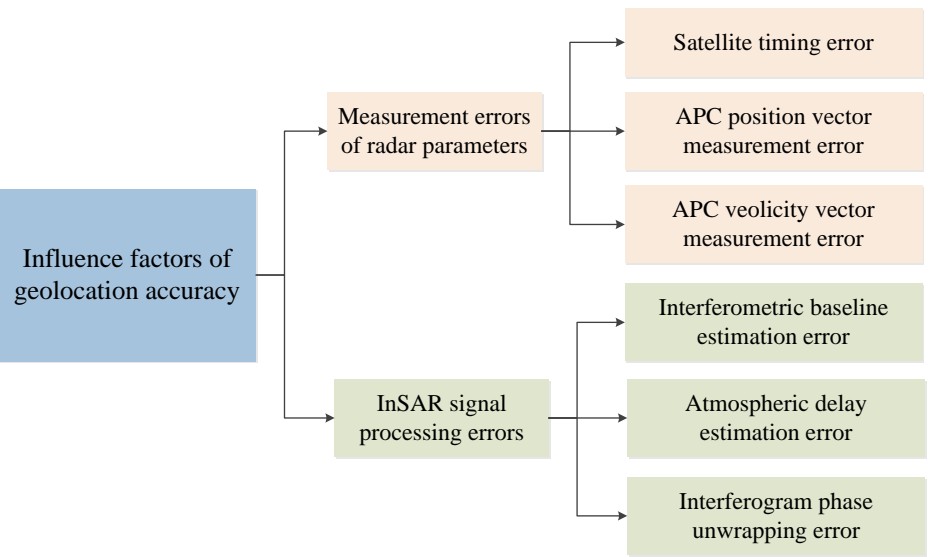

**Figure 4.** Influencing factors of geolocation accuracy.

### 4. Methodology

In this section, descriptions of the methods used in this paper are presented, including the ICESat-2 data filtering method, the baseline calibration method, the IRD geolocation method, and the low-coupling parallel calculation method.

### 4.1. ICESat-2 Data Filtering Method

The ICESat-2 data download file includes detailed parameter specifications, which record the quality information of the data, including meteorological information such as cloud thickness and terrain slope angle, etc. This quality information can be used as an important reference when conducting data filtering. If the cloud thickness is more than 3 m or the terrain slope angle is greater than 20 degrees, the data measurement accuracy of that day is determined to be low, and all sample points in that area are filtered out.

The ICESat-2 data are further filtered utilizing the AW3D DSM and World DEM datasets as reference datasets. The AW3D DSM dataset provides a publicly available digital surface model (DSM) with a spatial resolution of 30 m, similar to the ICESat-2 data. The elevation measurements of the AW3D DSM dataset are better than 7 m in areas with terrain slopes less than 30 degrees and better than 12 m in areas with terrain slopes greater than 30 degrees. The World DEM dataset also provides a publicly available digital elevation model (DEM) with a spatial resolution of 30 m. This DEM exhibits a measurement accuracy of better than 2 m in areas characterized by terrain slopes below 30 degrees and better than 5 m in regions with terrain slopes greater than or equal to 30 degrees.

The above two reference datasets can be used as the basis for screening ICESat-2 data. If the elevation measurement of ICESat-2 data differs from the two reference elevation data by more than 10 m (the threshold can be adjusted according to the specific situation) at the same latitude and longitude coordinates, it is considered an abnormal measurement sample point.

The data filtering process of ICESat-2 data is illustrated by the flowchart in Figure 5. In the flowchart, $H_{ALT}$ represents the elevation of the ICESat-2 samples, $H_{AW3D}$ represents the elevation of the AW3D DSM samples, and $H_{DEM}$ represents the elevation of the World DEM samples.

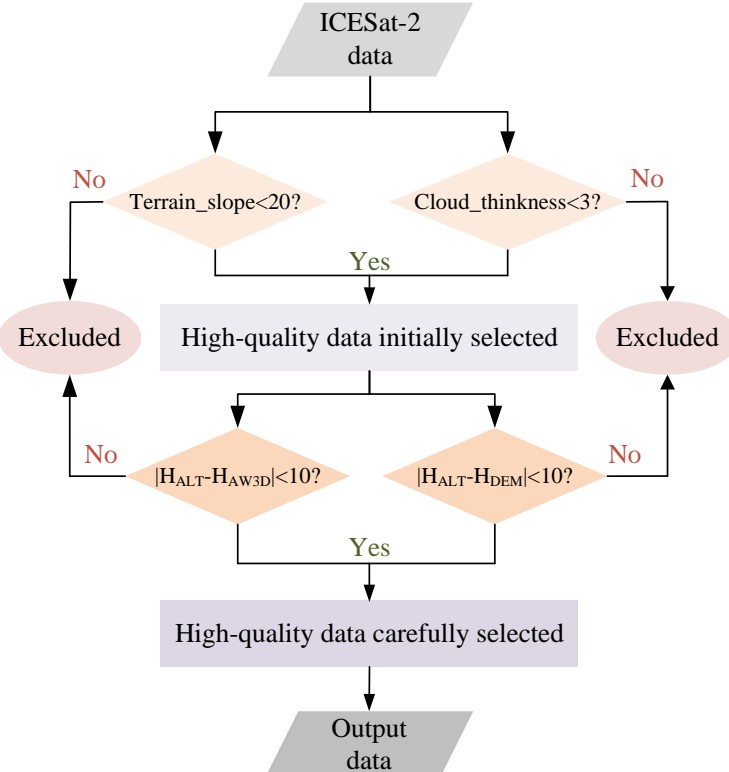

**Figure 5.** The flowchart for filtering ICESat-2 altimetry data.

### 4.2. Baseline Calibration Method

**Step 1: Coregistration of SAR image data and ICESat-2/GCP data**

The ICESat-2/GCP data record the geographic coordinates of each sample point $T_k$, and its correspondence with the pixel coordinates ($line_k$, $sample_k$) in a SAR image can be determined by applying Equation (21)–(23).

$$\frac{(\mathbf{T}_k - \mathbf{S}_{AT}(t_k)) \cdot \mathbf{V}_{AT}(t_k)}{\lambda \|\mathbf{T}_k - \mathbf{S}_{AT}(t_k)\|} + \frac{(\mathbf{T}_k - \mathbf{S}_{AR}(t_k + \Delta t)) \cdot \mathbf{V}_{AR}(t_k + \Delta t)}{\lambda \|\mathbf{T}_k - \mathbf{S}_{AR}(t_k + \Delta t)\|} = 0 \tag{21}$$

where $t_k$ represents the zero-Doppler moment of the sample point, and there is a correspondence between $t_k$ and the SAR image start time $t_{start}$, as shown in Equation (22). The symbol $\Delta t$ in Equation (21) represents the propagation time of radar signals from transmission to reception, which can be calculated using Equation (23).

$$t_k = t_{start} + line_k \cdot PRT \tag{22}$$

PRT in Equation (22) represents the pulse repetition time interval. The values of PRT and $t_{start}$ are recorded in the meta.xml file accompanying the SAR images.

$$\Delta t = \frac{R_{near} + PixelSpace \cdot sample_k}{c} \tag{23}$$

where $R_{near}$ denotes the slant range of the first pixel in each line, and *PixelSpace* represents the sampling interval between adjacent pixels along the range dimension. Both parameters are measured in meters and are recorded in detail in the meta.xml file accompanying the SAR image. The symbol c represents the propagation velocity of electromagnetic waves.

**Step 2: Calculation of elevation measurement error**

The calculation method of the elevation measurement error can be divided into two cases based on the type of reference data. If the reference data are the GCP data with a similar resolution to the SAR image, the elevation measurement error can be directly obtained by subtracting the DEM elevation of the matching point from the GCP elevation, as shown in Equation (24).

$$h_{err} = h_{GCP} - h_{DEM} \tag{24}$$

For the second case, the correlation between the ICESat-2 reference data and experimental data is complex. The SAR image resolution is 3m× 3m, whereas the ICESat-2 laser footprint diameter is about 17.5m. According to the resolution ratio relationship, an ICESat-2 laser footprint contains about 25 SAR image pixels. The average elevation of these 25 SAR image pixels is calculated and subsequently compared with the ICESat-2 laser altimetry data to derive the elevation measurement error [32], as shown in Equation (25).

$$h_{err} = h_{ICESat-2} - \frac{\sum_{k=1}^{M} h_{DEM\_k}}{M} \tag{25}$$

where M represents the number of SAR image pixels within the ICESat-2 laser footprint, and $h_{DEM_k}$ represents the elevation measurement value of the k-th pixel.

Figure 6 exhibits a model diagram, where the red circle depicts the contour line of the ICESat-2 laser footprint, and the red dot in the center represents the center of the footprint. The solid green circles represent the locations of SAR image pixels within the footprint.

**Step 3: Estimation of parallel baseline error**

The parallel baseline error $B_{\|err}$ is parallel to the line of sight (LoS) and exhibits a linear relationship with the elevation measurement error $h_{err}$, as shown in Equation (26).

$$B_{\|err} = h_{err} \cdot \frac{B_\perp}{R_{AT} \sin(\theta_i)} \tag{26}$$

where $B_\perp$ represents the vertical baseline length, $R_{AT}$ represents the distance between the ICESat-2/GCP point and the satellite $S_{AT}$, and $\theta_i$ represents the incidence angle of the corresponding pixel in the SAR image.

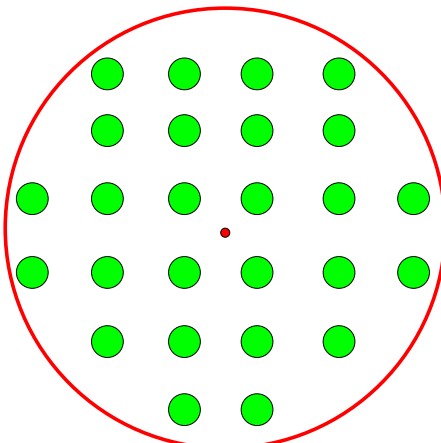

**Figure 6.** The correlation between the ICESat data and experimental data.

**Step 4: Estimation of perpendicular baseline error**

The interferometric baseline error is a kind of systematic error that varies slowly over time. At any given moment, the baseline error vector can be decomposed into two orthogonal components: the parallel baseline error $\mathbf{B}_{\|err}$ and the perpendicular baseline error $\mathbf{B}_{\perp err}$. $\mathbf{B}_{\|err}$ is parallel to the LoS, and $\mathbf{B}_{\perp err}$ is perpendicular to the LoS. The direction of the LoS varies from near range to far range, so the directions of $\mathbf{B}_{\|err}$ and $\mathbf{B}_{\perp err}$ are also dynamic for different range samples. In Figure 7, the decomposition result of the baseline error vector $\Delta\mathbf{B}_s$ at a near-range sample comprises $\mathbf{B}_{\|err1}$ and $\mathbf{B}_{\perp err1}$, whereas the corresponding result at a far-range sample consists of $\mathbf{B}_{\|err2}$ and $\mathbf{B}_{\perp err2}$.

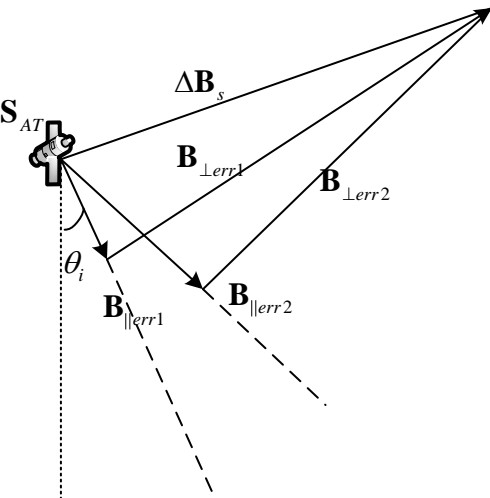

**Figure 7.** Interferometric baseline error vector decomposition diagram.

Based on the equality of the vector synthesis results obtained from these two decomposition methods, an equation can be established, as shown in Equation (27).

$$\mathbf{B}_{\|err1} + \mathbf{B}_{\perp err1} = \mathbf{B}_{\|err2} + \mathbf{B}_{\perp err2} = \Delta\mathbf{B}_s \tag{27}$$

Then, the estimation of the perpendicular baseline error can be determined via the least-squares fitting method.

**Step 5: Baseline error coordinate system conversion**

According to the relationship between the baseline error and radar incidence angle, the expression of the baseline error in the satellite's local coordinate system can be deduced, as shown in Equations (28)–(30).

$$x_s = B_\perp \cos(\theta_i) + B_\parallel \sin(\theta_i) \tag{28}$$

$$z_s = B_\parallel \cos(\theta_i) - B_\perp \sin(\theta_i) \tag{29}$$

$$\Delta\mathbf{B}_s = (x_s, 0, z_s)^T \tag{30}$$

**Step 6: Baseline error correction**

The baseline measurement error estimated using the baseline calibration method is typically defined in the satellite's local coordinate system, whereas the satellite orbital data (including position and velocity vectors) is defined in the Earth-centered Earth-fixed (ECEF) coordinate system. Therefore, it is necessary to convert the baseline error estimation from the satellite's local coordinate system into the ECEF coordinate system and subsequently integrate it with the original satellite orbital data. The satellite position vector $\mathbf{S_{AT}}$ and velocity vector $\mathbf{V_{AT}}$ in the coordinate transformation formula are associated with the azimuth time of pixels, so it is impossible to carry out baseline correction for the entire scene based on the baseline error expression of reference points. The formula for coordinate transformation from the satellite's local coordinate system to the ECEF coordinate system is shown in Equations (31)–(33).

$$\mathbf{T}_x = -\mathbf{S}_{AT}(t_{azi}) \times \mathbf{V}_{AT}(t_{azi}) \tag{31}$$

where $\mathbf{T}_x$ represents the transformation expression of the x-axis from the satellite's local coordinate system to the ECEF coordinate system, where the symbol "$\times$" represents the cross-product of two vectors. $\mathbf{S_{AT}(t_{azi})}$ and $\mathbf{V_{AT}(t_{azi})}$ represent the satellite's position vector and velocity at time $t_{azi}$, respectively.

$$\mathbf{T}_z = \mathbf{T}_x \times \mathbf{V}_{AT}(t_{azi}) \tag{32}$$

where $\mathbf{T}_z$ represents the transformation expression of the z-axis from the satellite's local coordinate system to the ECEF coordinate system.

$$\Delta\mathbf{B}_E = x_s \cdot \frac{\mathbf{T}_x}{\|\mathbf{T}_x\|} + z_s \cdot \frac{\mathbf{T}_z}{\|\mathbf{T}_z\|} \tag{33}$$

where $\Delta\mathbf{B}_E$ represents the baseline error vector in the ECEF coordinate system, and $\Delta\mathbf{B}_S = (x_s, 0, z_s)^T$ represents the baseline error vector in the satellite's local coordinate system.

*4.3. IRD Geolocation Method*

This subsection introduces the flowchart of the proposed IRD geolocation algorithm, as illustrated in Figure 8. The whole process can be divided into four steps. Here, the algorithm processing steps are described in detail.

**Step 1: Data pre-processing**

This step includes orbital interpolation processing and baseline calibration. The original orbital sampling points are sparse, usually one record per second. However, the two adjacent pixels in SAR images are sampled more densely than this. The acquisition of precise measurements of the orbital position and velocity at the moment of ground target focusing necessitates the implementation of orbital interpolation processing. Additionally, baseline calibration is also completed in this step. It is worth noting that the estimated baseline calibration value is a constant number in the local coordinate system, but it becomes a variable number after projection into the Earth-centered fixed coordinate system.

Therefore, a pixel-wise correction of the satellite's position is imperative during the orbital interpolation process.

**Step 2: Establishment of geolocation equation system**

The geolocation equation system comprises two range equations and one Doppler equation. The input parameters of the range equation include the satellite transmitter position $\mathbf{S}_{AT}$ and receiver position $\mathbf{S}_{AR}/\mathbf{S}_{PR}$ corresponding to the zero-Doppler moment of ground target $\mathbf{P}_T$. These can be obtained through the orbital interpolation process in Step 1. Additionally, the slant range between the satellite and ground target can be estimated based on the delay time from signal transmitting to receiving. Assuming that the satellite $\mathbf{S}_{AT}$ sends a signal at time $t_0$ and receives it at time $t_1$, the propagation time of the electromagnetic wave is $t_1 - t_0$. The propagation path of electromagnetic waves during this period can be divided into two distinct segments: the transmission distance $R_{AT}$ and the reception distance $R_{AR}$. The sum range $\hat{R}_{ATA}$ consists of $R_{AT}$ and $R_{AR}$, which can be estimated using Equation (34).

$$\hat{R}_{ATA} = R_{AT} + R_{AR} = c \cdot (t_1 - t_0) \tag{34}$$

where $c$ represents the propagation speed of electromagnetic waves in a vacuum; however, the propagation speed of electromagnetic waves in the atmosphere is slower than that in a vacuum. The estimated value of $\hat{R}_{ATA}$ according to Equation (34) is larger than the true value. Thus, range calibration is needed. In the other range equation, the sum range of $R_{AT}$ and $R_{PR}$ is calculated based on the interferometric phase $\varphi$ and slant range $\hat{R}_{ATA}$, as shown in Equation (35).

$$\hat{R}_{ATP} = R_{AT} + R_{PR} = \hat{R}_{ATA} + \frac{\varphi}{2\pi}\lambda \tag{35}$$

The relative values of $\hat{R}_{ATP}$ and $\hat{R}_{ATA}$ in Equation (35) are determined with exceptional precision, because the measurement sensitivity of the interference phase method is higher than that of the time delay method.

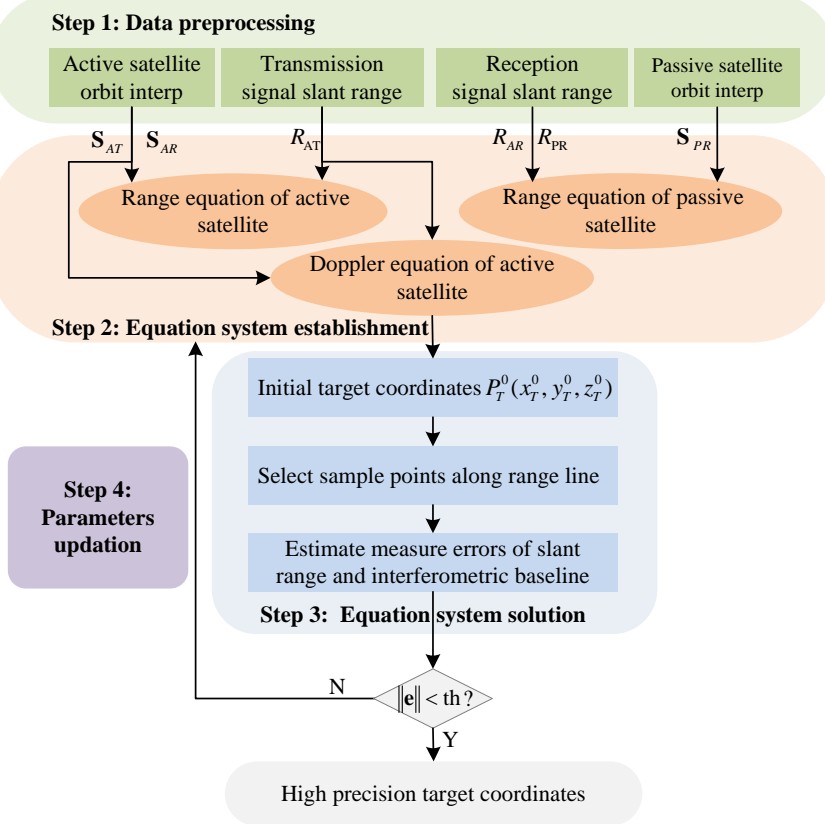

**Figure 8.** Flowchart of BiInSAR target geolocation algorithm.

**Step 3: Equation system solution**

The Newton iteration method is utilized to solve the nonlinear geolocation equations, aiming to minimize the standard deviation of the errors. First, an initial value of $P_T^0 = (x_T^0, y_T^0, z_T^0)^T$ is estimated based on the pixel coordinates $(azi, rgi)$ of $P_T$ and the radar parameters. Then, the initial value $P_T^0$ is substituted into Equation (19) to calculate the iterative correction vector **u**. The relative error is calculated by selecting sample points along the range line to ensure the consistency of all ground targets. The ICESat-2 laser points are matched to the experimental SAR data, enabling the accurate estimation of range errors caused by atmospheric propagation delays. These range errors seriously reduce the consistency of target geolocation accuracy.

**Step 4: Accuracy evaluation and parameter updating**

The geolocation deviation vector **e** and iterative correction vector **u** can be obtained according to Equation (19). The vector **e** contains evaluation information about the range deviation and Doppler center frequency deviation. If both the range deviation and Doppler center frequency deviation are below the threshold, the target geolocation equation convergence and we obtain the output coordinate $\hat{\mathbf{P}}_T^k = (x_T^k, y_T^k, z_T^k)^T$. Otherwise, if the range deviation or the Doppler center frequency deviation is higher than the threshold, the estimation of $\hat{\mathbf{P}}_T^k$ is not accurate enough. A new estimation value is given by $\hat{\mathbf{P}}_T^{k+1} = \hat{\mathbf{P}}_T^k + \mathbf{u}$. If the iterations reach the upper limit and do not converge, it means that the input parameters of Equation (19) have significant errors. Therefore, it is necessary to correct errors such as baseline vector, atmosphere delay, etc.

### 4.4. Low-Coupling Parallel Calculation Method

The low-coupling parallel calculation method aims to decompose the time-consuming computational tasks in the algorithm and assign these tasks to different threads. Each thread independently acquires the necessary data from shared memory, performs data processing, and ultimately returns the processed results to the main thread. The entire processing procedure consists of three fundamental stages: data preparation, task allocation, and parallel processing, as illustrated in Figure 9.

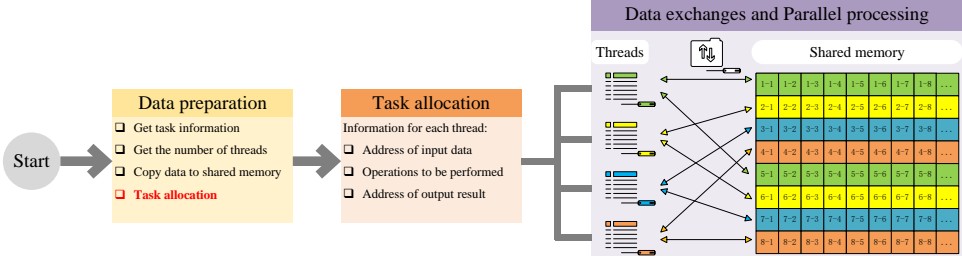

**Figure 9.** A flowchart of the parallel processing method.

**Step 1: Data preparation**

Data preparation mainly involves obtaining task information, determining the number of threads available for the task, copying input data into shared memory, and assigning tasks according to the number of threads.

**Step 2: Task allocation**

Task allocation includes assigning the input and output data addresses for each thread, as well as specifying the operations to be performed.

**Step 3: Parallel processing**

After receiving the task, each thread engages in data interaction with the shared memory and parallel processing. In Figure 9, four distinct color-coded thread symbols (green, yellow, blue, and orange) are employed to represent four independent computing units. The rectangular block on the right signifies the data stored in shared memory. The data correlation between each thread and memory is achieved by employing a consistent

color scheme. The data required by each thread for processing can be distributed in the shared memory either continuously or discretely, depending on the actual demand.

## 5. Experimental Design and Analysis of Results

In this section, two groups of experiments are designed to verify the proposed IRD geolocation algorithm. It is proven that the proposed algorithm overcomes the problem of poor precision in monostatic equivalent (MoE) algorithms [21] and exhibits a faster iteration speed compared to the existing entropy minimize principle (EMP) algorithm.

### 5.1. Group 1: Digital Simulation Experiment

This subsection presents a digital simulation experiment, and the simulation parameters are listed in Table 1. The simulated scene comprises nine precisely georeferenced point targets. The effectiveness of the proposed algorithm is verified through three distinct test conditions. The first test takes place under ideal conditions, devoid of any measurement errors, and yields geolocation accuracy on the order of 0.001 m. In contrast, the second test takes place under challenging conditions, marked by multiple measurement errors and the absence of calibration. In this scenario, geolocation accuracy is on the order of 5 m. For the third test, multiple measurement errors are introduced, but calibration processes are incorporated, resulting in geolocation accuracy on the order of 0.1 m. These experiments offer valuable insights into the algorithm's performance under varying conditions, from ideal to real-world scenarios, providing a comprehensive assessment of its capabilities.

**Table 1.** Simulation experiment parameters of BiInSAR strip imaging mode.

| Satellite Parameter | Value |
| --- | --- |
| Semi-major axis | 6913.140 km |
| Orbital altitude | 535.00 km |
| Inclination | 97.54° |
| Eccentricity | 0 |
| Off-nadir angle | 41.19° |
| Radar frequency | 9.2 GHz |
| Transmitted bandwidth | 150 MHz |
| Sampling rate | 180 MHz |
| Pulse repetition frequency | 6,763 Hz |
| Transmitter velocity | 7681.69 m/s |
| Antenna size (azimuth × range) | 5 m × 3 m |

The nine point targets are distributed at approximately equal intervals in the range and azimuth dimensions, as displayed in Figure 10. The objective of this distribution design is to explore the potential correlation between the geolocation error distributions and pixel coordinates, encompassing azimuth and range coordinates. The baseline measurement errors include the parallel baseline error and perpendicular baseline error. In this experiment, the baseline measurement error $\Delta \vec{B}_e = \left( b_1 \cdot \vec{B}_{\parallel} + b_2 \cdot \vec{B}_{\perp} \right)$ is a constant value, and the values of $b_1$ and $b_2$ are listed in Table 2. The value of the slant-range measurement error $\Delta R_e$ is affected by cloud thickness and terrain height. In this experiment, the value of $\Delta R_e$ is set according to the exponential model and terrain height, as shown in Equation (36).

$$\Delta R_e(h, \theta_{inc}) = \frac{ZPD}{\cos(\theta_{inc})} \cdot e^{-\frac{h}{H}} \tag{36}$$

where the Zenith Path Delay (ZPD) is an empirical parameter, fixed at a constant 2.3 m for an X-band signal. The reference height $H$ is set to 6000 m, and $h$ is the scene average height, which can be extracted from a coarse DEM. The types and values of measurement errors are listed in Table 2.

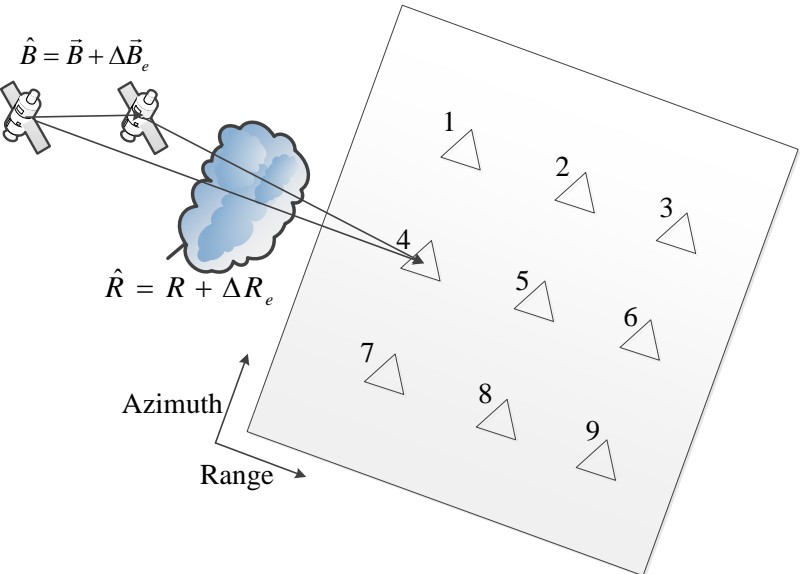

**Figure 10.** Design of simulation experiment.

**Table 2.** Types and values of measurement errors in the simulation experiment.

| Parameter Type | Parameter Name | Measurement Error Value |
|---|---|---|
| Radar parameters | Satellite timing error | 0.0015 s |
| | APC position vector measurement error | 0.5 m |
| | APC velocity vector measurement error | 0.01 m/s |
| Signal processing parameters | Parallel baseline error | 0.002 m |
| | Perpendicular baseline error | 0.002 m |
| | Atmospheric delay estimation error | 0.2–0.8 m |
| | Interferometric phase unwrapping error | 3° |

The geolocation accuracies of point targets under different test conditions are depicted in Figure 11. The red line represents the geolocation error trend of the nine point targets under ideal conditions (no measurement errors). The blue line represents the geolocation error trend under bad conditions with the baseline measurement error. The orange line represents the geolocation error trend under worse conditions with both the baseline and range errors. The purple line represents the geolocation error trend under good conditions with the baseline measurement error and carrying out parameter calibration. The pink line represents the geolocation error trend under good conditions with multiple types of measurement errors and carrying out parameter calibration. A partially enlarged image of the curves at the bottom of Figure 11 is provided on the right of the original image.

Through this series of comparative experiments, it was found that the geolocation accuracy was mainly affected by two factors. One factor was the number of error types and the magnitude of measurement errors, and the other was whether parameter calibration was carried out. The measurement accuracy of the slant range based on the time delay method was around 1 m. In our method, the measurement accuracy can reach the same order as the carrier wavelength, usually at the decimeter or even centimeter level. The root-mean-square error (RMSE) is used to reflect the accuracy of target geolocation. It can be seen that even if there are measurement errors, as long as the magnitude of errors does not exceed the threshold, the proposed method can effectively estimate the values of the measurement errors. By carrying out parameter calibration, the geolocation accuracy can reach a similar level to the accuracy under ideal conditions, as exhibited in Table 3.

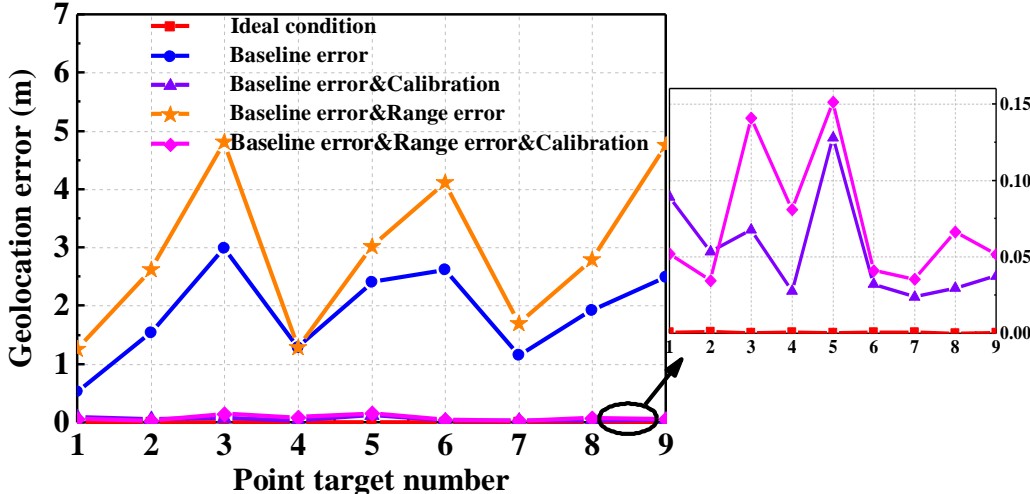

**Figure 11.** Geolocation accuracy of point targets under different test conditions.

**Table 3.** RMSEs of target geolocation under different test conditions.

| Conditions | Number of Error Types | Parameter Calibration | RMSE of Target Geolocation |
|---|---|---|---|
| Ideal | Zero | - | 0.001 m |
| Bad | One | No | 2.45 m |
| Worse | Two or more | No | 3.21 m |
| Good | One | Yes | 0.08 m |
| Good | Two or more | Yes | 0.11 m |

The geolocation errors of the point targets were evaluated, and the comparison results of the different algorithms are exhibited in Figure 12. The proposed IRD algorithm exhibits a lower average error and smaller standard deviation compared to the MoE and EMP algorithms. The experimental results depicted in Figure 12 demonstrate the absence of a significant correlation between geolocation errors and pixel coordinates.

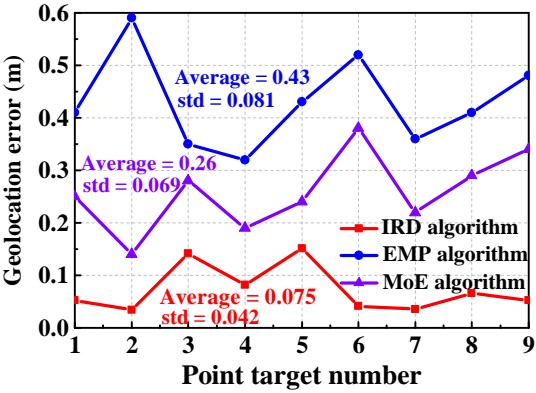

**Figure 12.** A comparison of geolocation errors across different algorithms.

Based on the existing experiments, a group of incremental baseline errors with a step of 1cm was added to the experimental data. Subsequently, an estimation value for the baseline error was obtained using the proposed algorithm and evaluated based on the residual error. The test results are exhibited in Figure 13, where the horizontal axis represents the baseline error introduced in the experiment, and the vertical axis represents the residual error after conducting baseline calibration. The curve in Figure 13 demonstrates that the residual baseline errors exhibit a fluctuation trend centered around zero means, with amplitudes

smaller than 0.5 mm under a boundary condition where the baseline measurement error does not exceed 20 cm.

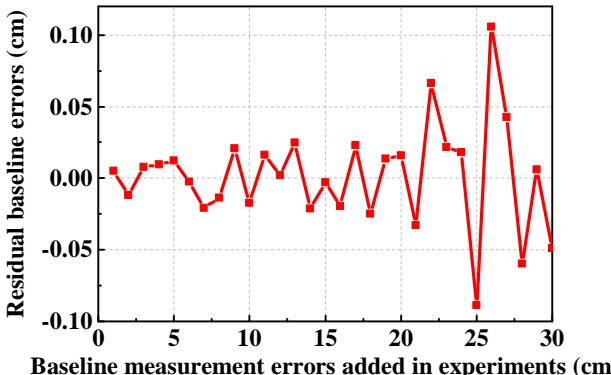

**Figure 13.** The boundary condition experiment for the baseline error.

### 5.2. Group 2: Real SAR Data Experiment

In this subsection, two typical kinds of bistatic SAR satellite data are used as examples to verify the efficiency of the proposed geolocation algorithm in DEM generation applications.

The first test data were acquired by the TH2-01A and TH2-01B satellites on 26 September 2019. The key parameters of this test data are listed in Table 4. The TH-2 satellites and the whole radar system were designed and built in China and launched on 30 April 2019. The imaging mode of the test data is a strip map, and both the azimuth resolution and range resolution are 3 m. The amplitude image and coherence image of the TH-2 test data are exhibited in Figure 14.

**Table 4.** Key parameters of TH-2 test data.

| Parameter Name | Parameter Value |
| --- | --- |
| Satellite name | TH2-01A, TH2-01B |
| Orbital height | 580 Km |
| Incidence angle | 42.1° |
| Nearest range | 600 Km |
| Resolution | 3 m |
| Perpendicular baseline length | 280 m |
| Height of ambiguity | 21 m |
| Average coherence | 0.91 |

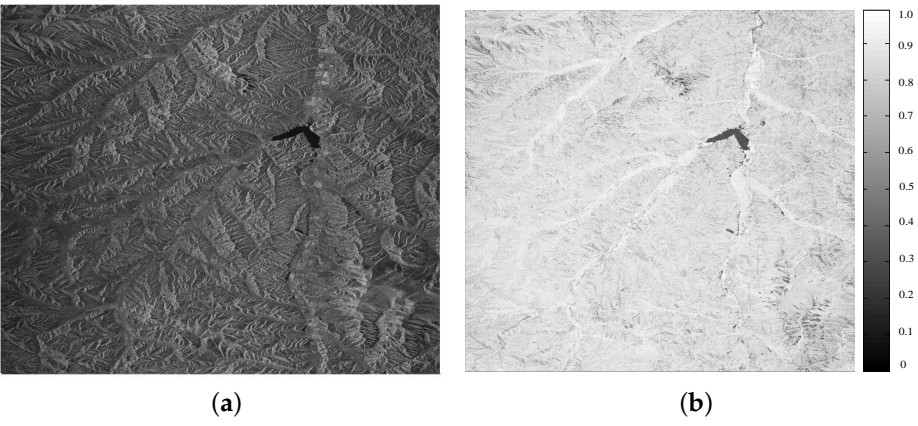

**Figure 14.** Amplitude image and coherence image of TH-2 test data. (**a**) Amplitude information. (**b**) Coherence information.

The size of the original SAR image is 23,716 × 23,596 pixels. After conducting the necessary pre-processing tasks for the SAR images, the efficiency of the proposed IRD geolocation algorithm can be verified. Data pre-processing includes image coregistration, interferometric phase extraction, phase filtering, and phase unwrapping. These pre-processing jobs can be performed using various commercial software, such as Gamma, ENVI SARscape, RDSpace, PIE, SAR studio, etc. These data pre-processing tasks are applicable to both monostatic satellite data and bistatic satellite data. The primary focus of this paper is on geolocation processing.

There are 10 trihedron reflectors in the test image, and their geographical coordinates can be obtained via a high-precision GPS device. The GPS device exhibits a measurement accuracy at the level of 0.5 m, rendering its measurement values suitable as reference benchmarks for evaluating the precision of diverse geolocation algorithms. The distribution of the 10 GCPs is illustrated in Figure 15. The red squares in Figure 15 represent the pixel coordinates of the GCPs in the SAR image, and at the end of each red connecting line is a partially enlarged image. Figure 15 is defined in the slant-range coordinate system, where the horizontal axis represents the slant-range dimension, and the vertical axis represents the azimuth dimension.

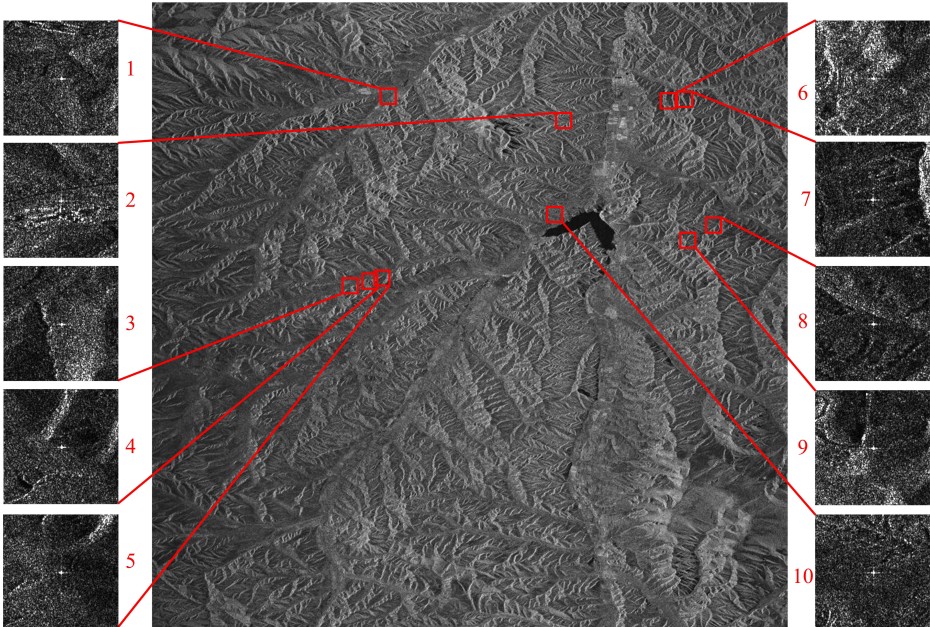

**Figure 15.** Pixel coordinates of GCPs and their partially enlarged images.

The proposed IRD geolocation algorithm was utilized to compute geographical coordinates for all pixels within the SAR image, subsequently enabling the generation of a three-dimensional topographic map of the observed scene, as depicted in Figure 16. The red dots represent the positions of the GCPs, and the numbers near the red dots represent the order of the GCPs. Figure 16 is defined in the geographical coordinate system, where the horizontal axis represents the longitude coordinate, and the vertical axis represents the latitude coordinate. Since the distributions of the GCPs are discrete and sparse, ICESat-2 laser altimetry data are provided as reference data to evaluate the geolocation accuracy of the proposed method. The distribution of ICESat-2 laser altimetry data is illustrated in Figure 17, where the white dots represent the geographic positions of laser points.

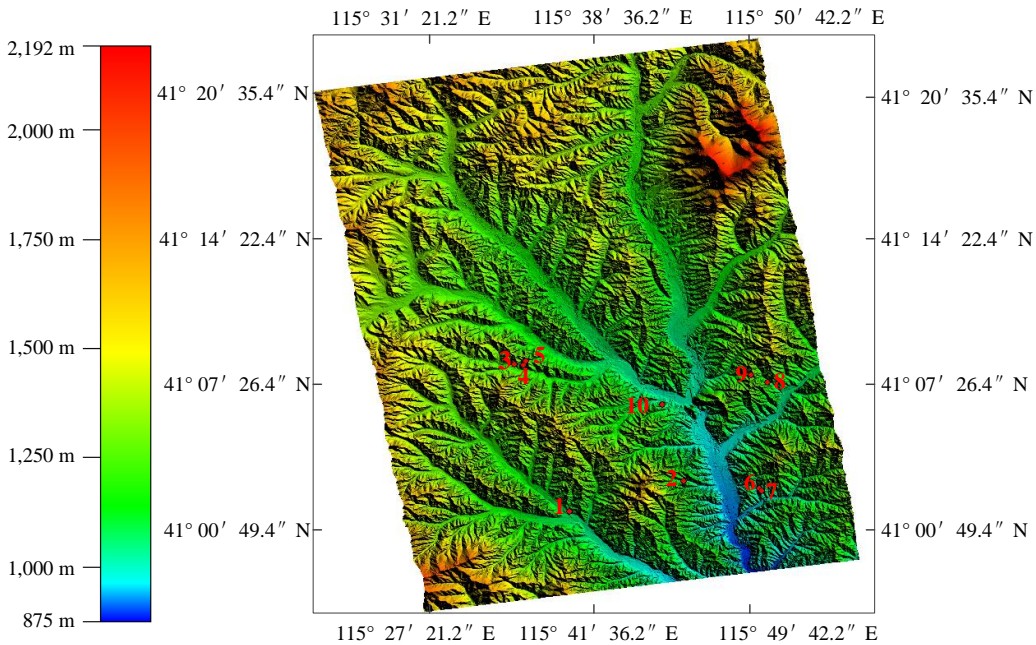

**Figure 16.** Three- dimensional topographic map of the geolocation results.

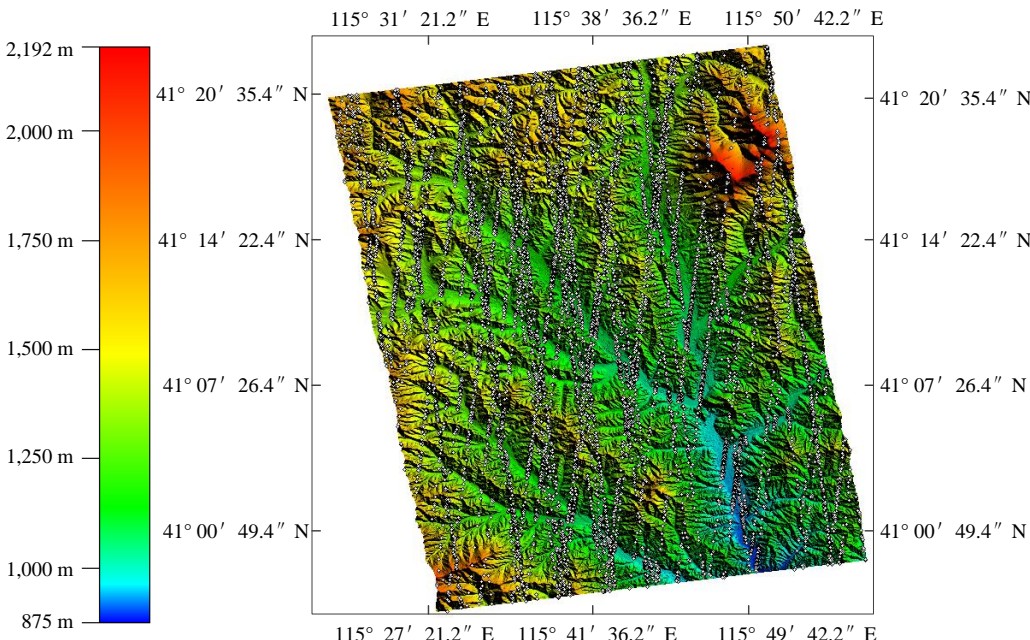

**Figure 17.** ICESat-2 laser altimetry data distribution.

The results depicted in Figure 18 demonstrate that the proposed IRD algorithm exhibits geolocation errors smaller than those observed with the EMP and MoE algorithms, as evidenced by both the GCP reference data and the ICESat-2 laser altimetry data. In Figure 18a, the horizontal axis represents the order of the GCPs, and the vertical axis represents the geolocation error of each point. In Figure 18b, the horizontal axis represents the statistics intervals of elevation error, with a range of [−15 m, 15 m] and an interval of 0.5 m. The vertical axis represents the number of pixels in each interval. It can be seen that the average elevation error of the IRD algorithm is closer to zero and its standard deviation(std) is smaller than the other two algorithms.

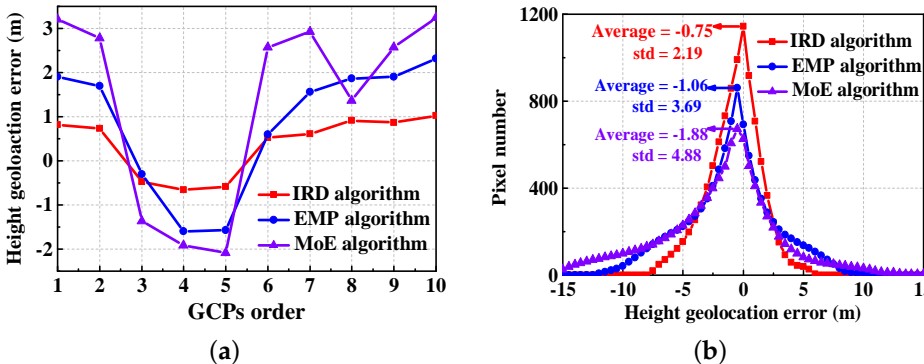

**Figure 18.** Evaluation of height geolocation accuracy. (**a**) Evaluation results based on GCPs. (**b**) Evaluation results based on ICESat-2 laser altimetry data.

In Figure 18, we can see that both the GCP reference data and the ICESat-2 laser altimetry data prove that the geolocation errors of the proposed IRD algorithm are smaller than those of the EMP and MoE algorithms. The evaluated RMSE of the IRD algorithm based on the GCP reference data is 0.743 m, whereas the RMSEs of the EMP and MoE algorithms are 0.778 m and 2.49 m, respectively. And, the RMSEs of the three algorithms based on the ICESat laser altimetry data are 3.190 m, 3.439 m, and 5.234 m, respectively. The reason why the RMSEs evaluated based on ICESat-2 laser altimetry data are larger than the RMSEs evaluated based on the GCP reference data is that the ICESat-2 laser altimetry measurement values represent the average height values for each footprint of the laser, whereas the GCP measurement values represent specific point heights. The TH-2 test image coverage is in a mountainous area, and the average height (ICESat-2 data) does not exactly match the geolocation product, so the evaluation result is larger than the real geolocation error.

In Figure 18a, the geolocation error curves show depressions at control points 3, 4, and 5. The local terrain slope angle of this area is statistically analyzed to assist in investigating the cause. The local terrain slope map of the generated digital elevation model (DEM) is presented in Figure 19a, with the positions of the control points denoted by red dots. The color bar on the right side of Figure 19a represents the degrees of the terrain slope angle. Additionally, Figure 19b illustrates a one-dimensional terrain slope profile along the longitude direction for three selected control points. It can be seen that the variation trends for the terrain slope angle around control points 3, 4, and 5 are at critical points of rising and falling. Due to the unique topography, these three control points exhibit distinct geolocation errors compared to the remaining control points. The geolocation errors demonstrate a correlation with the pixel coordinates, which arises from the differences in the terrain height and/or slope angle of the GCPs.

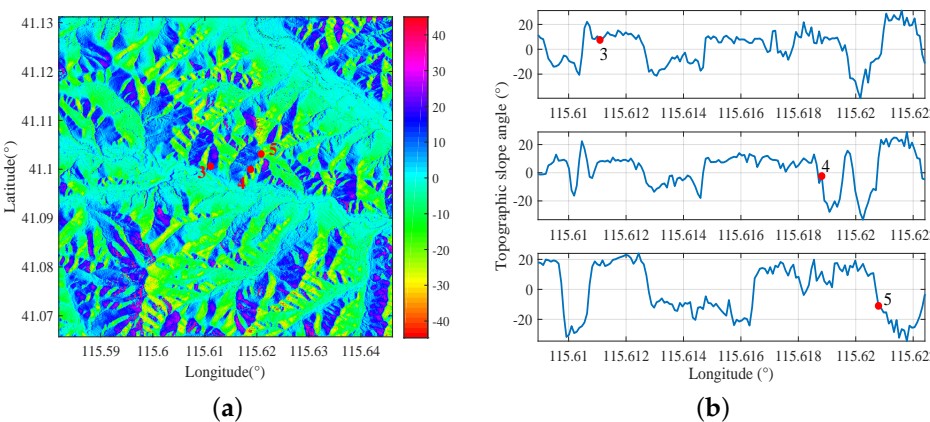

**Figure 19.** The local terrain slope angle of DEM. (**a**) Two-dimensional view. (**b**) One-dimensional view.

*5.3. Algorithm Efficiency Improvement*

In this subsection, the efficiency of the proposed low-coupling parallel calculation method is verified. It can improve the calculation speed by two to three times. The core idea is to reduce the coupling between computing units and improve computing efficiency via the CPU-based parallel computing strategy. The creation of a shared memory block enables universal access by all threads. Subsequently, individual data processing is performed in parallel by each thread, culminating in the submission of the processed results to the main thread. When designing parallel computing algorithms, special consideration should be given to ensuring that the memory addresses of the local variables defined by each thread are unique and that there are no memory conflicts. For example, the absolute phase matrix and the phase-to-elevation conversion matrix are defined as shared memory that can be accessed by all threads, whereas the satellite orbit position vector, velocity vector, and slant-range variables are defined as local variables. Algorithm designers allocate computing tasks reasonably among various computing units, thereby improving computational efficiency.

Table 5 shows the processing times for different geographic positioning algorithms under the same experimental conditions. The computer used in our experiment has an Intel (R) Core (TM) i7-4900 processor running @ 3.2 GHz, with eight threads and 32 GB of memory. The experimental data include two sets of TH-2 data and two sets of TerraSAR-X data. The average size of the experimental data is $18848 \times 26045$ pixels, covering an area of around 30 km$^2$. It can be seen that regardless of the geolocation accuracy, the IRD algorithm processes around 1,429,678 pixels per second, whereas the EMP algorithm processes about 674,157 pixels per second. This proves that the proposed IRD algorithm requires less time than the EMP algorithm to process the same test data.

**Table 5.** The processing times of different geolocation algorithms.

| Information on Test Data | | Data Processing Times | |
|---|---|---|---|
| Test Data ID | Test Data Size | EMP Algorithm | IRD Algorithm |
| TH2-01BA-InSAR-20190926 | $21{,}096 \times 23{,}584$ pixels | 12.3 min | **5.8 min** |
| TH2-01AB-InSAR-20191015 | $23{,}716 \times 23{,}596$ pixels | 14.25 min | **6.65 min** |
| TDM1-SAR-BIST-SM-20180223 | $13{,}206 \times 28{,}796$ pixels | 9.3 min | **4.13 min** |
| TDM1-SAR-BIST-SM-20130101 | $17{,}374 \times 28{,}204$ pixels | 11.2 min | **5.3 min** |

## 6. Conclusions

This paper aims to improve geolocation accuracy and calculation efficiency in target geolocation. Firstly, a bistatic interferometric baseline calibration model and an improved range–Doppler (IRD) model are proposed. These two models effectively reflect the geometric characteristics in a bistatic configuration. Then, the calculation steps and derivation process of the baseline calibration method and the IRD target geolocation method are elaborated in detail. A processing flowchart is provided to introduce how the parallel processing method proposed in this paper improves the efficiency of the algorithm. The proposed algorithm is verified using a pair of bistatic spaceborne SAR data acquired by the TH-2 satellite, which is China's first spaceborne BiInSAR system. Compared with the MoE and EMP algorithms, the accuracies of the proposed IRD algorithm can be improved by around 80 percent and 40 percent, respectively. Meanwhile, compared with the EMP algorithm, the calculation efficiency is improved by two to three times by reducing the coupling between computation units and designing a parallel calculation strategy. In conclusion, the proposed BiInSAR geolocation algorithm based on the IRD model has been successfully employed in the DEM generation and topographic mapping mission of the TH-2 satellite.

**Author Contributions:** Conceptualization, C.X. and F.T. (Fanyi Tang); methodology, F.T. (Feng Tian); software, C.X.; validation, C.X.; investigation, C.X.; writing—original draft preparation, C.X.; writing—review and editing, Z.L., F.T. (Fanyi Tang), F.T. (Feng Tian), and Z.S.; funding acquisition, Z.L. and Z.S. All authors have read and agreed to the published version of the manuscript.

**Funding:** This research was funded by the National Natural Science Foundation of China (grant no. 62031005).

**Data Availability Statement:** Not applicable.

**Acknowledgments:** The authors would like to thank the anonymous reviewers for their helpful comments and suggestions.

**Conflicts of Interest:** The authors declare no conflicts of interest.

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
