# Peer review of "A High-Precision Target Geolocation Algorithm for a Spaceborne Bistatic Interferometric Synthetic Aperture Radar System Based on an Improved Range–Doppler Model"

_remotesensing, doi:10.3390/rs16030532_

Round 1
Reviewer 1 Report
Comments and Suggestions for Authors
The·authors propose a high-precision and·efficient geolocation algorithm for a BilnSAR system·based·on·an·improved RD·model. The effectiveness of the algorithm is·demonstrated through·the experimental data acquired by the TH-2 satellite, which is the first spaceborne BilnSAR system of China. However, additional effort is needed. Here are my comments:
1.·The English of the manuscript still needs some improvement, if it is possible, I would suggest a native speaker to read over the text. For example:
(1) In P2, the expression “carry wavelength” is not consistent with the expression “carrier wavelength” in P4 L3.
(2) In P8, Subsection 3.2, the tense is inconsistent in sentence “Assume satellite SAT transmitted signal at time t0, and received signal at time t1, the sum range RATA consists of RAT and RAR”.
(3) In P10, Section 4, the citation format of reference [21] is inconsistent with other references.
2. Compared with Eq.8, the Eq.1 and Eq.2 are incomplete. The authors need to check carefully.
3. In Fig.1, there is a yellow line between SAT and SPR, but the authors said that “The black connect line between SAT and SPR represents the instantaneous baseline (ISB)”. The authors need to confirm.
4. In Fig.7, the legend is ambiguous. What does the single error and multi-errors stand for? Besides, the curves in the bottom of Fig.7 are not clear, a partial enlarge image is needed.
5. Why are the distributions of GCPs in Fig.9 and Fig.10 different?
6. In Fig.12(a), why are the geolocation error curves sunken at control points 3 to 5?
7. In Fig.12(b), there are too many points, making the picture information difficult to read. Could you provide some statistical information such as average value, standard deviation to highlight the superiority of the algorithm.
Reviewer 2 Report
Comments and Suggestions for Authors
Minor corrections required.
Comments on the Quality of English LanguageDear Authors,
The idea of using ICESat-2 as the input / validation data is good. However its filtering and related important aspects are missing. The study are is a hilly region, so GCPs / ICESat-2 usage shall be clearly mentioned and described. Other suggestions are:
1. The results of the presented study (if/in-case any, as evident form the statement: The measurement accuracy of the slant range based on the time delay method is around 1 meter.) shall be moved to “Results and Discussion” Section.
2. Page 3: As mentioned in: “The black connect line between SAT and SPR represents the instantaneous baseline (ISB)….” In figure Yellow and green line are visible? Kindly clarify or correct for the referred Black line.
3. The write up on Figure 1 shall be made clear to meke the deduction of relation clear as mentioned on Page 2 (Section 2, first paragraph).
4. How the variability in the ICESat-2 data uncertainity is dealt? The terrain and canopy height uncertainity columns in the ICESat-2 data has not been discussed. How they are used, shall be made clear in the methodology.
5. Methodology Sections shall be provided separately before the “Results and Discussion” Section. Currently, its mixed with section 3 on “BiInSAR Geolocation Algorithm Based on IRD Model”. Its suggested to use standard manuscript format.
6. Conclusion section shall be rewritten in the light of above changes.
Reviewer 3 Report
Comments and Suggestions for Authors
-
This paper presents an advanced geolocation algorithm designed for a spaceborne bistatic interferometric Synthetic Aperture Radar (BiInSAR) system, employing an enhanced Improved Range-Doppler (IRD) model to achieve both high precision and operational efficiency. The manuscript is well-crafted, with adept organization; however, a few minor adjustments are required.
(1)In Figure 12-a, it is imperative not to omit Ground Control Points (GCP) numbers 1, 3, 5, 7, and 9.
-
(2)Section 4.3 lacks a comprehensive elucidation of the principles underlying the enhancement of algorithm efficiency.
Reviewer 4 Report
Comments and Suggestions for Authors
The paper introduces a high-precision and efficient geographic positioning algorithm for a spaceborne bistatic interferometric SAR system based on an improved range Doppler (IRD) model. A comprehensive description of the spatial baseline geometric model unique to the dual base configuration was provided, and the vertical baseline expression was derived; Establish an IRD geographic positioning function to meet the specific requirements of dual base configuration. On this basis, a bistatic interferometric SAR positioning algorithm based on the IRD function is proposed. By modifying the distance Doppler equation to meet the dual base configuration, the accuracy of target localization can be significantly improved. At the same time, a low coupling merging row calculation method was proposed to improve computational efficiency. The accuracy and effectiveness of the algorithm have been verified by simulation and measured data.
Revision suggestions:
In the explanation of Figure 1, there is a mismatch between the colors in the text and the colors in the figure
2. In section 3.2 on page 8, step 3, and sections 3.3 on page 9, if laser data correction is required for each point, it seems that the work is meaningless. If reference point error is used as the global error for correction, it will cause inaccurate correction of non reference points. The author suggests using GCPs or ICESat2 laser altimetry data as reference data to calibrate interferometric baseline errors, The paper should clearly indicate and provide a specific corrective process.
On page 8, section 3.2, step 4, and page 12, for the explanation of Table 3, the proposed algorithm can estimate the value of the error well under a certain threshold, and corresponding boundary conditions should be considered.
4. The color in Figure 12 (b) is closer to blue and black, and the comparison image is not clear enough. Changing to a color with stronger contrast would be more reasonable.
5. The author proposes a low coupling merging row calculation method that utilizes shared memory and parallelization to optimize the computational efficiency of the proposed IRD algorithm. Can all EMP algorithms used as a comparison apply this optimization approach? An explanation should be provided that the proposed method can be parallelized while the EMP algorithm cannot.
Comments on the Quality of English LanguageMinor editing of English language required.
Round 2
Reviewer 2 Report
Comments and Suggestions for Authors
Results shall be added to the abstract. Make abstract as per standard requirements with all required elements.
Validation for simulation shall be improved/detailed. (Compared with the MoE and EMP algorithms, the accuracies of the proposed IRD algorithm can be improved by around 80 percent and 40 percent, respectively.)
9-point method and 10 GCPs shall be logically placed/ supported by reasoning / benefits.
Comments on the Quality of English Language
Third person tense suggested with completed sentences
